# Synthesis and Properties of Norphthalocyanines Functionalized with a Tetrathiacrown Ether–Tetrathiafulvalene Substituent

**DOI:** 10.3390/molecules28030916

**Published:** 2023-01-17

**Authors:** Ruibin Hou, Xiaoyu Liu, Yan Xia, Dongfeng Li

**Affiliations:** 1School of Chemistry and Life Science, Changchun University of Technology, Changchun 130012, China; 2Advanced Institute of Materials Science, Changchun University of Technology, Changchun 130012, China

**Keywords:** tetrathiafulvalene, norphthalocyanine, tetrathiacrown ether, intramolecular charge-transfer, density functional theory

## Abstract

To construct novel ion receptors and D-A self-assembly systems for materials with better functions, the annulation of a tetrathiafulvalene donor with a magnesium norphthalocyanine core via a flexible tetrathiacrown ether bridge afforded a new triad **1**. The structure of this product was characterized by 1H NMR and infrared spectroscopy, time-of-flight mass spectrometry, and elemental analysis. The optical and electrochemical properties were investigated using UV–vis spectroscopy and cyclic voltammetry. The complex of triad **1** and 2,3,5,6- tetrafluoro-7,7,8,8-tetracyanoquinodimethane produced electron transfer with a radical cationic character, as confirmed by UV–Vis and electron paramagnetic resonance analysis. Furthermore, the target compound presented evident intramolecular charge-transfer interactions in ground states, which were explained using density functional theory. Furthermore, norphthalocyanine **1** was able to coordinate Ag^+^ through the peripheral ligating oxathiaether crown.

## 1. Introduction

Silver is a naturally occurring transition metal and a noble metal. It is known to be discharged to the environment from its industrial applications, especially in its ionic form (Ag^+^), which is detrimental to aquatic and terrestrial organisms and is known to convert into the more innocuous forms as it reacts rapidly with natural chemical ligands in sewer systems, sewage treatment, and the environment [1].

The formation of supramolecular sensors based on a self-assembly strategy for molecular and ion recognition has become a hot research field. Recently, the construction of ensembles composed of covalently linked electron-donor (D) and electron-acceptor (A) moieties is important in the fields of materials science and supramolecular chemistry. Usually, well-designed D or A building blocks with a wide range of functional units can rationally control the unique optical and electrical properties of D-A materials. The performance of D-A compounds is controlled by three intrinsic factors, namely, the natures of the donor, the acceptor, and their connecting units [2,3,4,5,6]. The typical D-A ensembles are the polyads, which are derived from phthalocyanine, porphyrin, porphyrazine with macrocyclic tetrapyrroles moiety as the A part and TTFs as the D part. Investigating electronic interactions in such systems is useful to understand electron transfer in D-A molecules. Phthalocyanine (Pc) derivatives directly annulated with four TTF units or substituted with four or eight TTF units at peripheral positions have previously been prepared and their electron-transfer behavior investigated. Accordingly, TTF-modified Pc has a mainly symmetric structure [7,8,9,10,11]. Recently, we reported unsymmetrical Pzs (norphthalocyanines) bearing TTF units, including the norphthalocyanines (NPc) annulated directly with one TTF moiety using crown ether as a linker and linked two TTF units through ethylenedithio spacers [12,13]. These NPc bearing TTF units showed good intramolecular charge transfer (ICT) properties that were clearly dependent on the nature of the linkage between the redox-active units.

To construct novel ion receptors and D-σ-A self-assembly systems for materials with better functions, we described the synthesis and photophysical and electrochemical properties of NPc derivative **1**. Compound **1** was composed of one TTF unit as the electron donor and a single Mg-norphthalocyanine ring as the electron acceptor, covalently linked by a tetrathia-18-crown-6-ether spacer. The structure of the TTF moiety improved the complexation ability of the tetrathia-18-crown-6-ether with cations through an electron-donating process.

## 2. Results and Discussion

### 2.1. Synthesis and Quantum Chemical Calculations

Key precursor TTF-dicyanotetrathiacrown ether **2** was obtained according to previously reported procedures [14]. As shown in Figure 1, the mixed condensation of 1,2-dicyanobenzene (20 equiv) and derivative **2** (1 equiv) under classic Linstead macrocyclization conditions produced two phthalocyanine products, namely, the desired mono tetrathiafulvalene–tetrathiacrown ether Mg–norphthalocyanine **1** and Mg-phthalocyanine. Fortunately, differences in the solubility and polarity of these two products allowed their easy separation to afford the desired norphthalocyanine **1**. The MALDI-TOF mass spectrum of **1** featured a peak at *m*/*z* 1247.85 [M + H]^+^, corresponding to the M^+^ ion (1246.21) of **1**. The 1H NMR spectrum of **1**, recorded in CDCl_3_ at 24 °C, showed signals attributed to the CH_2_O and CH_2_S protons in the crown ether units and alkyl groups linked to the TTF skeleton, which were characteristic of the proposed structure. All experimental procedures and characteristic data are detailed in the Appendix A.

To gain deeper insight into the molecular and electronic structures, mono TTF-NPc **1** was further examined using theoretical calculations. The molecular geometry was optimized at the B3LYP/6-31G** level of theory. All calculations were performed using the Gaussian 09 package [15]. The frontier molecular orbitals are shown in Figure 1. In compound **1**, the highest occupied molecular orbital (HOMO) was mainly located on the TTF moiety, while the lowest unoccupied molecular orbital (LUMO) was mainly located on the Mg–norphthalocyanine ring. These data were in good agreement with the expected electron−donor and electron−acceptor properties, acting as a prerequisite to the intramolecular charge transfer (ICT) transition and positive shift in the oxidation potentials of the donor subunit [16,17].

### 2.2. Electrochemical and Photophysical Properties

To evaluate the potential of the target compound to act as an electron donor, electrochemical characterization of compound **1** in CH_3_CN–CH_2_Cl_2_ (1:5, *v*/*v*) was performed by cyclic voltammetry (CV). The cyclic and differential pulse voltammograms of **1** within a potential window of −1.8 V to +1.8 V are shown in Figure 2. Three oxidation couples (E_1/2_ = 0.439 V, 0.895 V, and 1.466 V) and two reduction couples (E_1/2_ = −1.431 V and −1.081 V) were observed within this potential window in the CH_3_CN–CH_2_Cl_2_/Bu_4_PF_6_ electrolyte system. These five couples were assigned to TTF^+^/TTF (I), TTF^+2^/TTF^+^ (II), NPc-1/NPc-2 (III), NPc-3/NPc-4 (IV), and NPc-2/NPc-3 (V) on the basis of previous reports. Processes III and V were irreversible in terms of the ratio of anodic to cathodic peak currents. In contrast, processes I, II, and IV, which were assigned to simultaneous first and second oxidations of the TTF unit, and second reduction of the NPc unit, respectively, were quasi-reversible [18,19,20].

The optical spectra of Mg-NPc **1** exhibited two main bands; a Soret or B band (π-π*, corresponding to a deep π-LUMO transition) at 300–400 nm and another at approximately 600–820 nm, denoted as the Q band (π-π*), as shown in Figure 3. In general, the four-fold symmetric phthalocyanine macrocycles generated similar optical spectra, with single transitions for both the Q and B bands. In contrast, the optical spectra of unsymmetrical Mg-NPc **1**, comprising three fused benzo rings and one TTF unit substituted with a thiacrown ether moiety, clearly showed a splitting of the Q band. These optical spectra resembled those reported in the literature for NPcs substituted with 2,3-dialkylthio groups [21]. These differences were explained using Gouterman’s highly simplified four-orbital model for the optical spectra of tetrapyrrole macrocycles. To further address the donor properties of the newly synthesized compounds, doping studies were conducted using 7,7,8,8-tetracyanoquinodimethane (TCNQ) and 2,3,5,6-tetrafluoro-7,7,8,8-tetracyanoquinodimethane (F_4_TCNQ) in CH_2_Cl_2_. For Mg-NPc **1** doped with TCNQ (2 equiv) in CH_2_Cl_2_; no CT bands were observed in the 600–1000 nm region. However, when Mg-NPc **1** was doped with F_4_TCNQ (1 equiv), a new absorption band was produced at approximately 865 nm in the UV-Vis spectrum (Figure 3). This new band corresponded to the cation radical species of the TTF moieties. The formation of the F_4_TCNQ^−^/TTF^+^ charge-transfer complex in CH_2_Cl_2_ was also confirmed by electron paramagnetic resonance (EPR) spectroscopy. The EPR spectrum of Mg-NPc **1**, recorded below 25 °C, showed resonances centered at approximately *g* = 2.008 and 2.002, which were regions characteristic of a TTF radical cation [22] and an F_4_TCNQ radical anion [23], respectively. These results indicated that some CT occurred between the TTF unit and F_4_TCNQ in the solution (Figure 4).

The influence of metal ions on key precursor **2** and Mg-NPc **1** was investigated by absorption titration. As thiacrown ethers are known to coordinate heavy metal ions, the metal-binding studies of **1** focused on the complexation of heavy metal ion Ag^+^ [24,25]. The addition of AgClO_4_ to a solution of **2** in CH_2_Cl_2_-MeOH (7:3, *v*/*v*) led to changes in the absorption spectra of the ligands, namely, increased intensity and red-shifting of the absorbances at approximately 230, 321, and 443 nm, and decreased intensity and blue-shifting of the absorbance at approximately 332 nm (Figure 5). The appearance of two isosbestic points at approximately 326 and 410 nm indicated the formation of well-defined Ag^+^-binding complexes.

The titration of Mg-NPc **1** against AgClO_4_ is shown in Figure 6. Ag^+^ binding showed marked effects on the macrocycle B and Q bands. Adding AgClO_4_ initially caused a blue shift and sharpening of the Q band up to a metal/**1** ratio of approximately 1:1, while the B bands showed a red shift and enhanced intensity (Figure 6). The addition of perchlorate cations other than silver perchlorate caused no change in the absorption spectra of **1**. Notably, no change in the absorption spectra was observed upon the addition of various other alkali metal ions, such as Li^+^, K^+^, and Cs^+^ as shown in Appendix A, indicating the good complexation of **1** and **2** for Ag^+^. Meanwhile, the absorption spectra were checked upon the addition of soft transition metal ion Zn^2+^ as described in Appendix A. The result demonstrated that no change was found. However, we did not check the 12-membered crown for encapsulation of Ag^+^. We think the cavity of the 12-membered crown is small if it could encapsulate Ag^+^; it could form a sandwich compound.

## 3. Experimental Section

### Instruments and Chemicals

NMR spectra were recorded in CDCl_3_ with a Bruker AV-300 Spectrometer and chemical shifts were referenced relative to tetramethylsilane (δ_H_/δ_c_ = 0). IR spectra were recorded on a Perkin Elmer 2400 instrument (Waltham, MA, USA, KBr pressed disc method). MALDI-TOF-MS data were obtained by a Shimadzu AXIMA-CFRTM (Kyoto, Japan) spectrometer, using a 1, 8, 9-anthracenetriol (DITH) matrix. Ultraviolet-visible (UV-vis)spectra were recorded on a L 25 spectrophotometer in CH_2_Cl_2_. Cyclic voltammetric studies were carried out using a CHI852C instrument (Shanghai, China) with CH_3_CN-CH_2_Cl_2_ as the solvent (10^–3^ M) and 0.1 MBu_4_ClO_4_ as the supporting electrolyte. Counter and working electrodes consisted of a Pt wire and a Pt disk, respectively, and the reference electrode was a calomel electrode (SCE). All the reagents were purchased from J&K Technology Co., Ltd (Beijing, China). Key precursor **2** was synthesized according to the literature method [14]. The molecular geometry was optimized by using the B3LYP/6-31G** level of theory. All calculations were performed utilizing the Gaussian 09 package.

*Synthesis of Mg-norphthalocyanine* (**1**): Magnesium (16.3 mg, 0.68 mmol) metal was dissolved in anhydrous *n*-BuOH (45 mL) at reflux under Ar. To this magnesium butoxide solution, compound **2** (100 mg, 0.23 mmol) and phthalonitrile (589 mg, 4.6 mmol) were added. The mixture was refluxed for 26 h under Ar. The solution changed from purplish red to deep blue. The blue mixture was cooled to room temperature. The precipitate was collected by suction and washed with a large amount of CHCl_3_. The blue solid was purified by chromatography on silica gel with CH_2_Cl_2_/CH_3_OH (200:1–20:1, *v*/*v*) to give **1** as a deep blue solid (5.5 mg, yield 8.6%). Reprecipitation of **1** from CH_2_Cl_2_–MeOH gave a deep blue powder (melting point >250 °C by differential thermal analysis).

1H NMR (CDCl_3_): *δ* (ppm) = 0.85 (br, 6H), 1.24 (br, 16H), 1.38 (br, 4H), 1.63 (br, 4H), 2.81 (br, 4H), 2.97 (br, 4H), 3.08 (br, 12H), 7.73 (br, 2H), 7.83 (br, 4H), 8.35 (br, 2H), 8.89 (br, 4H); FT-IR (cm^−1^): 2920.23, 2848.86, 1540.21, 1501.63, 1334.74, 1302.82, 1116.78, 1020.34, 887.26, 738.74, 723.31, 669.30; MALDI-TOF MS *m*/*z* = 1247.85 (M^+^, 100%, calcd. 1278.12).

## 4. Conclusions

A new TTF-tetrathiacrown ether-NPc triad (**1**) with two long and flexible aliphatic side chains at the periphery was prepared and characterized. Theoretical calculations provided an improved understanding of the electronic structures of the interacting molecules. The ability of compound **1** to function as a donor for F_4_TCNQ was established, and evidence for the formation of a charge-transfer complex was obtained by UV-vis and EPR spectroscopy. The presence of 1 equiv of AgClO_4_ showed marked effects on the macrocycle B and Q bands of **1**, indicating that **1** would be a good Ag^+^ sensor. The synthesis and properties of larger tetrathiacrown ether analogs are currently being investigated.

## Data Availability

Data related to this study is available from the corresponding author upon reasonable request.

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
