# Peer review of "Synthesis and Properties of Norphthalocyanines Functionalized with a Tetrathiacrown Ether–Tetrathiafulvalene Substituent"

_molecules, 2023, doi:10.3390/molecules28030916_

Round 1

Reviewer 1 Report (Previous Reviewer 3)

The manuscript (molecules-2113563) is the resubmission of the previous manuscript (molecules-1856177) with the same title from Li et al.

1. Page 5, last paragraph. The author claimed ‘The addition of perchlorate cations other than silver perchlorate caused no change in the absorption spectra of 1.’ What I understand is, several perchlorate salts (usually more than ten) were tested. So, I was asking what are those perchlorate salts before. But based on the response from the author to my third comment, it seems only LiClO4, KClO4 and CsClO4 were tested. (I’m even not sure if these three salts are the ones that were tested since the counter ion for the three alkali metal ion were never mentioned anywhere) I cannot agree with the conclusion that compound 1 could ‘selectively’ bind silver by testing only 4 ions, especially the other three are all alkali metal ions. The experimental results are far from enough to support the conclusion of ‘good complexation selectivity of 1 and 2 for Ag+’. At least, ions from different groups need to be tested with both compounds.

2. The last question of my previous comments was not responded to: For the 1H-NMR spectrum of compound 1, why the integration is so different between the number of protons from the structure and the spectrum? If the author explains with ‘perhaps this complex is easy to aggregate to obtain broad peak, so not easy to ascription peak’, some evidence showing the reason for the aggregation should be added. Or it will be hard for me to believe the compound is pure. The mass spectrum could not prove the purity of the compound.

Author Response

  1. Page 5, last paragraph. The author claimed ‘The addition of perchlorate cations other than silver perchlorate caused no change in the absorption spectra of 1.’ What I understand is, several perchlorate salts (usually more than ten) were tested. So, I was asking what are those perchlorate salts before. But based on the response from the author to my third comment, it seems only LiClO4, KClO4and CsClO4were tested. (I’m even not sure if these three salts are the ones that were tested since the counter ion for the three alkali metal ion were never mentioned anywhere) I cannot agree with the conclusion that compound 1 could ‘selectively’ bind silver by testing only 4 ions, especially the other three are all alkali metal ions. The experimental results are far from enough to support the conclusion of ‘good complexation selectivity of 1 and 2 for Ag+’. At least, ions from different groups need to be tested with both compounds.

Response: we think this suggestion is good. we think this three alkali metal ion are representative and with good solubility. For counter ion we never think this compound with crown ether structure could response. So we didn’t check this aspect of experimental.   

  1. The last question of my previous comments was not responded to: For the 1H-NMR spectrum of compound 1, why the integration is so different between the number of protons from the structure and the spectrum? If the author explains with ‘perhaps this complex is easy to aggregate to obtain broad peak, so not easy to ascription peak’, some evidence showing the reason for the aggregation should be added. Or it will be hard for me to believe the compound is pure. The mass spectrum could not prove the purity of the compound.

Response: This similar phenomenon has been reported in phthalocyanines. We didn’t know what other evidence to explain the aggregation. This is not purity problem for the compound.

Reviewer 2 Report (Previous Reviewer 2)

The responses to my review are very appropriate, I think. 

The authors said, "we added other soft transition metal for encapsulating property in supporting information."  Figure S7 is very convincing. However, there is no description of this figure in this manuscript. An explanation for Figure S7 should be added.

To my question, "whether did encapsulation into 12-membered crown for Ag+ occur or not?",  the authors answered, "we didn’t check 12-membered crown for encapsulation of Ag+. We think the cavity of 12-membered crown is small, if it could encapsulate Ag+, it could form a sandwich compound." This description had better be added to this manuscript, I think.

It was expected that the interpretation for red-shift and enhanced intensity of B band could discuss further in this manuscript,  but if the authors have no idea at the present time, it can't be helped.

Author Response

The authors said, "we added other soft transition metal for encapsulating property in supporting information."  Figure S7 is very convincing. However, there is no description of this figure in this manuscript. An explanation for Figure S7 should be added.

Response: we have added explanation for Figure S7 in manuscript as red color.

To my question, "whether did encapsulation into 12-membered crown for Ag+ occur or not?",  the authors answered, "we didn’t check 12-membered crown for encapsulation of Ag+. We think the cavity of 12-membered crown is small, if it could encapsulate Ag+, it could form a sandwich compound." This description had better be added to this manuscript, I think.

Response: we have added this description in manuscript.

It was expected that the interpretation for red-shift and enhanced intensity of B band could discuss further in this manuscript, but if the authors have no idea at the present time, it can't be helped.

Response: thank you for this suggestion, but now we have no idea and perhaps later could discuss further in this manuscript.

This manuscript is a resubmission of an earlier submission. The following is a list of the peer review reports and author responses from that submission.

Round 1

Reviewer 1 Report

The manuscript is generally well written, but I have a few concerns:

1) Overall significance of the work is not well highlighted. Please improve introduction with clear goals of project and rationale behind it.

2) Is the new band at 865 nm observed in the absorption spectra for Mg-NPc-1 doped with F4TCNQ truly a new band or shifted band to the one seen around 820 nm in 1?

3) Paragraph on effect of AgClO4 titration with Mg-NPc 1 is not clear. How does adding Ag affect the observed changes in the spectra of Mg-NPc 1?

4) Where is the data for the titrations with other alkali metal ions? A common figure should be included to truly highlight the selectivity.

5) The data included in supplementary files is not acceptable. Proper electronic files of all NMR and other data should be included. There should be traceability to the included data. Cell phone images of data is not acceptable for publication.

Author Response

Reviewer 1:

1) Overall significance of the work is not well highlighted. Please improve introduction with clear goals of project and rationale behind it.

Response: we have improved the introduction with clear goals of project as red color.

2) Is the new band at 865 nm observed in the absorption spectra for Mg-NPc-1 doped with F4TCNQ truly a new band or shifted band to the one seen around 820 nm in 1?

Response: this is new band at 865 nm observed in the absorption spectra.

3) Paragraph on effect of AgClO4 titration with Mg-NPc 1 is not clear. How does adding Ag affect the observed changes in the spectra of Mg-NPc 1?

Response: the paragraph on effect of AgClO4 titration with Mg-NPc 1 has been modified as red color. Adding Ag affected the observed changes in manuscript we have described.

4) Where is the data for the titrations with other alkali metal ions? A common figure should be included to truly highlight the selectivity.

Response: we added the titrations with some alkali metal ions experimental in supporting information. 

5) The data included in supplementary files is not acceptable. Proper electronic files of all NMR and other data should be included. There should be traceability to the included data. Cell phone images of data is not acceptable for publication.

Response: we try to make proper electronic files of NMR and other data.

Reviewer 2 Report

This manuscript is the review described about synthesis and characterization of norphthalocyanines functionalized with a tetrathiacrown ether and tetrathiafulvalene substituent. In addition, electron transfer behavior and encapsulating of metal ion into thiacrown ether has been explained. The system of reaction and obtained structures are are also characterized definitely.

In fact, encapsulating of silver ion in 18-membered ring has been explained in this manuscript. However, it is well-known that thiacrown ether with soft donor prefers silver ion and thus it does not have necessity that complex 2 molecules must be applicated, I think. (Actually, molecule 1 also showed encapsulation of a silver ion.) If the authors want to show the difference from those in ref.11, comparison of  encapsulation into 12-membered crown should been described and relationship between the encapsulation of metal ions and the charge transfer, which is main theme in the paper, behavior should be further discussed. Finally, the authors said,” 1 was a good candidate as a Ag+ sensor.” If say so, researches about encapsulating of other transition metal ions also should be carried out.

 Sorry, as mentioned above, this research only can be seen as one of routine work at the present stage, and thus it finds impossible for me to see this report has novelty.

Author Response

Reviewer 2:

This manuscript is the review described about synthesis and characterization of norphthalocyanines functionalized with a tetrathiacrown ether and tetrathiafulvalene substituent. In addition, electron transfer behavior and encapsulating of metal ion into thiacrown ether has been explained. The system of reaction and obtained structures are are also characterized definitely.

In fact, encapsulating of silver ion in 18-membered ring has been explained in this manuscript. However, it is well-known that thiacrown ether with soft donor prefers silver ion and thus it does not have necessity that complex 2 molecules must be applicated, I think. (Actually, molecule 1 also showed encapsulation of a silver ion.) If the authors want to show the difference from those in ref.11, comparison of  encapsulation into 12-membered crown should been described and relationship between the encapsulation of metal ions and the charge transfer, which is main theme in the paper, behavior should be further discussed. Finally, the authors said,” 1 was a good candidate as a Ag+ sensor.” If say so, researches about encapsulating of other transition metal ions also should be carried out. 

Sorry, as mentioned above, this research only can be seen as one of routine work at the present stage, and thus it finds impossible for me to see this report has novelty.

Response: thank you for your suggestion. This manuscript is not included researches about encapsulating of other transition metal ions. Later we continue to do this work and will be published results in other paper.

Reviewer 3 Report

The manuscript submitted by Li et al. presents the synthesis of a new unsymmetrical norphthalocyanines derivative characterized by 1H-NMR and mass spectrum. The photophysical and electrochemical properties of the new compound are studied by UV-vis, cyclic voltammetry and electron paramagnetic resonance. Furthermore, the coordination between cations and the new compound is studied with UV titration showing that only silver could be bound.

Similar norphthalocyanines derivatives have been studied by the author before (Ref. 11). The only difference between the two structures is the thiacrown ether linker. Exactly the same experiments as the previous work are carried out in the current one. The only new finding is the coordination with silver.

1. Did the author ever try the cation binding study with the compound in the previous work (Ref. 11)? If the compound in the previous work could also bind silver (selectively or not?), then what is the reason to make the new compound in this work? If it could not bind, it will be more convincing to synthesize the new compound. There definitely should be a connection between this work and the previous work. The author should explain more in the introduction section.

2. Again, compared with the previous work, the only new thing is binding silver selectively, why does this matter, and how this could be used in the future? A more detailed description needs to be added in the introduction section.

3. Page 5, line 132-139. In this paragraph, the binding selectivity of silver is described. But no specific cations are mentioned except silver. Also, ‘alkali metal ions, such as Li+, K+, and Cs+’ is not scientifically clear. Every cation that has been tested must be clarified clearly.

4. There are only 3 data points in the UV titration shown in figure 6. More data points should be added (referring figure 5).

5. Figure S1, the integration of the 1H-NMR spectrum could not match the analytical data reported in the manuscript.  

6. A 13C-NMR spectrum for the new compound should be added.

7. The supporting figures of 1H-NMR, mass spectrum and FT-IR should be changed to the electronic version.

Author Response

Reviewer 3:

The manuscript submitted by Li et al. presents the synthesis of a new unsymmetrical norphthalocyanines derivative characterized by 1H-NMR and mass spectrum. The photophysical and electrochemical properties of the new compound are studied by UV-vis, cyclic voltammetry and electron paramagnetic resonance. Furthermore, the coordination between cations and the new compound is studied with UV titration showing that only silver could be bound.

Similar norphthalocyanines derivatives have been studied by the author before (Ref. 11). The only difference between the two structures is the thiacrown ether linker. Exactly the same experiments as the previous work are carried out in the current one. The only new finding is the coordination with silver.

  1. Did the author ever try the cation binding study with the compound in the previous work (Ref. 11)? If the compound in the previous work could also bind silver (selectively or not?), then what is the reason to make the new compound in this work? If it could not bind, it will be more convincing to synthesize the new compound. There definitely should be a connection between this work and the previous work. The author should explain more in the introduction section.

Response: in Ref. 11 we did the selective experiment for compound and we make sure could not bind silver ion. So we did new compound in this manuscript for binding silver ion.

  1. Again, compared with the previous work, the only new thing is binding silver selectively, why does this matter, and how this could be used in the future? A more detailed description needs to be added in the introduction section.

Response: it is well-known that thiacrown ether with soft donor prefers silver ions.

  1. Page 5, line 132-139. In this paragraph, the binding selectivity of silver is described. But no specific cations are mentioned except silver. Also, ‘alkali metal ions, such as Li+, K+, and Cs+’ is not scientifically clear. Every cation that has been tested must be clarified clearly.

Response: we have added the alkali metal ions selectivity experimental in supporting information.

  1. There are only 3 data points in the UV titration shown in figure 6. More data points should be added (referring figure 5).

Response: we just choose 3 data points in the UV titration to demonstrate this complex can encapsulate silver ion. One reason is difficult to synthesize and purify this complex.

  1. Figure S1, the integration of the 1H-NMR spectrum could not match the analytical data reported in the manuscript.

Response: perhaps this complex is easy to aggregate to obtain broad peak, so not easy to ascription peak.

  1. 13C-NMR spectrum for the new compound should be added. 

Response: the quaternary carbon peak is not easy to appear in 13C-NMR spectrum, so we did not add the 13C-NMR spectrum.

  1. The supporting figures of 1H-NMR, mass spectrum and FT-IR should be changed to the electronic version.

Response: we have changed all the figures as electronic version.

Round 2

Reviewer 2 Report

Your response said, ”This manuscript is not included researches about encapsulating of other transition metal ions. Later we continue to do this work and will be published results in other paper.” However, if you want to say in this paper, “indicating the good complexation selectivity of 1 and 2 for Ag+” and “1 was a good candidate as a Ag+ sensor.”, comparison with phenomenon for the other soft transition metal is required here, not later. don't you think?

In addition, in previous review, I demanded in previous review explanation about the difference from phenomenon in ref.11, however,  there was no definite response about it. If you want to say this research is not "one of routine work", which only involved increase of the number of members in macrocycle and routine measurement of the compound, the difference should be described definitely in this paper. For example, whether did encapsulation into 12-membered crown for Ag+ occur or not? And so on.

Fortunately, question why happened red-shift and enhanced intensity of B band is much attractive, I think. The interpretation of this phenomenon should be discussed further in this paper, because the discussion was intimate for charge transfer of TTF-dicyanotetrathiacrown ether and encapsulating of metal ions.

 In conclusion, the revised manuscript has not yet reached little changing my opinion that this research only can be seen as one of routine work. Drastic revision is necessary, I think.

Reviewer 3 Report

It seems there are not many changes in the revised manuscript and supporting information. Here is the author's response to the comments I listed before and my new comment:

Similar norphthalocyanines derivatives have been studied by the author before (Ref. 11). The only difference between the two structures is the thiacrown ether linker. Exactly the same experiments as the previous work are carried out in the current one. The only new finding is the coordination with silver.

  1. Did the author ever try the cation binding study with the compound in the previous work (Ref. 11)? If the compound in the previous work could also bind silver (selectively or not?), then what is the reason to make the new compound in this work? If it could not bind, it will be more convincing to synthesize the new compound. There definitely should be a connection between this work and the previous work. The author should explain more in the introduction section.

Response: in Ref. 11 we did the selective experiment for compound and we make sure could not bind silver ion. So we did new compound in this manuscript for binding silver ion.

Reviewer: The introduction section is not sufficient to show the novelty and significance of this work. The only new function property of the newly synthesized compound is binding with silver ions by combining the naphthalocyanine ring with a thiacrown ether which is well known to be able to bind silver ions selectively. Although the author explains the compound from the previous work (Ref. 11) could not bind silver ions, there is no connection between the current and previous work or any results shown in the manuscript that could convince me of the significance of this work.

  1. Again, compared with the previous work, the only new thing is binding silver selectively, why does this matter, and how this could be used in the future? A more detailed description needs to be added in the introduction section.

Response: it is well-known that thiacrown ether with soft donor prefers silver ions.

Reviewer: I’m not asking why the new compound could bind the silver ions. I’m asking for why ‘binding silver ion’ or ‘complexation with soft metal cations’ is important. Why the ‘selectively’ is important? The publications should always be clear to wide audiences who may or may not know these. Again, the introduction section is not sufficient to show the significance of this work

  1. Page 5, line 132-139. In this paragraph, the binding selectivity of silver is described. But no specific cations are mentioned except silver. Also, ‘alkali metal ions, such as Li+, K+, and Cs+’ is not scientifically clear. Every cation that has been tested must be clarified clearly.

Response: we have added the alkali metal ions selectivity experimental in supporting information.

Reviewer: Page 5, line 134-141. ‘The addition of perchlorate cations other than silver perchlorate caused no change in the absorption spectra of 1’. What other perchlorate salts were tested? This is not a scientific expression. It should be clearly stated. No result of UV testing with other perchlorate salts is shown either in the manuscript or in the supporting information. For newly added figure S6, there are no experimental details like concentrations, what does ’ different amounts’ mean in the caption?

  1. Figure S1, the integration of the 1H-NMR spectrum could not match the analytical data reported in the manuscript.

Response: perhaps this complex is easy to aggregate to obtain broad peak, so not easy to ascription peak.

Reviewer: The 1H-NMR spectrum of a new compound reported in publications is a very important standard for not only showing the purity of the compound but also for other researchers to refer to in the future. It’s necessary to explain why the integration is so different between the number of protons from the structure and the spectrum. If the author explains with the above explanation, some evidence showing the reason for the aggregation should be added. Or it will be hard for me to believe the compound is pure. The mass spectrum could not prove the purity of the compound.